# SkinNet-INIO: Multiclass Skin Lesion Localization and Classification Using Fusion-Assisted Deep Neural Networks and Improved Nature-Inspired Optimization Algorithm

**DOI:** 10.3390/diagnostics13182869

**Published:** 2023-09-06

**Authors:** Muneezah Hussain, Muhammad Attique Khan, Robertas Damaševičius, Areej Alasiry, Mehrez Marzougui, Majed Alhaisoni, Anum Masood

**Affiliations:** 1Department of CS, HITEC University, Taxila 47080, Pakistan; 2Department of Computer Science and Mathematics, Lebanese American University, Beirut 13-5053, Lebanon; 3Department of Computer Science, HITEC University, Taxila 47080, Pakistan; 4Center of Excellence Forest 4.0, Faculty of Informatics, Kaunas University of Technology, 51368 Kaunas, Lithuania; robertas.damasevicius@ktu.lt; 5College of Computer Science, King Khalid University, Abha 61413, Saudi Arabia; areej.alasiry@kku.edu.sa (A.A.); mhrez@kku.edu.sa (M.M.); 6Computer Sciences Department, College of Computer and Information Sciences, Princess Nourah Bint Abdulrahman University, Riyadh 11564, Saudi Arabia; mmalhaisoni@pnu.edu.sa; 7Department of Circulation and Medical Imaging, Faculty of Medicine and Health Sciences, Norwegian University of Science and Technology (NTNU), 7034 Trondheim, Norway

**Keywords:** skin cancer, image processing, deep learning, features fusion, hyperparameters selection, feature selection, machine learning

## Abstract

**Background:** Using artificial intelligence (AI) with the concept of a deep learning-based automated computer-aided diagnosis (CAD) system has shown improved performance for skin lesion classification. Although deep convolutional neural networks (DCNNs) have significantly improved many image classification tasks, it is still difficult to accurately classify skin lesions because of a lack of training data, inter-class similarity, intra-class variation, and the inability to concentrate on semantically significant lesion parts. **Innovations:** To address these issues, we proposed an automated deep learning and best feature selection framework for multiclass skin lesion classification in dermoscopy images. The proposed framework performs a preprocessing step at the initial step for contrast enhancement using a new technique that is based on dark channel haze and top–bottom filtering. Three pre-trained deep learning models are fine-tuned in the next step and trained using the transfer learning concept. In the fine-tuning process, we added and removed a few additional layers to lessen the parameters and later selected the hyperparameters using a genetic algorithm (GA) instead of manual assignment. The purpose of hyperparameter selection using GA is to improve the learning performance. After that, the deeper layer is selected for each network and deep features are extracted. The extracted deep features are fused using a novel serial correlation-based approach. This technique reduces the feature vector length to the serial-based approach, but there is little redundant information. We proposed an improved anti-Lion optimization algorithm for the best feature selection to address this issue. The selected features are finally classified using machine learning algorithms. **Main Results:** The experimental process was conducted using two publicly available datasets, ISIC2018 and ISIC2019. Employing these datasets, we obtained an accuracy of 96.1 and 99.9%, respectively. Comparison was also conducted with state-of-the-art techniques and shows the proposed framework improved accuracy. **Conclusions:** The proposed framework successfully enhances the contrast of the cancer region. Moreover, the selection of hyperparameters using the automated techniques improved the learning process of the proposed framework. The proposed fusion and improved version of the selection process maintains the best accuracy and shorten the computational time.

## 1. Introduction

Skin cancer is one of the most prevalent cancers. For example, more than 5 million new cases are recorded annually in the United States, and it is anticipated that one in five persons may experience this illness at some point during their life [1]. It is a common malignancy that poses a major threat to human health and whose prevalence is rising annually around the globe [2]. Basal cell (BCC), squamous cell (SCC), and malignant melanoma are the most common skin malignancies, where the five year survival rate for BCC and SCC are above 95% [3]. Melanoma, a type of skin cancer, develops in the skin cells, and primarily is situated outside of the body, which is mainly exposed to ultraviolet rays from sunshine [4]. The World Health Organization estimates that 2–3 million new instances of skin cancer are diagnosed worldwide each year [5]. According to the facts, more than two people in the U.S. die with skin cancer every hour. In the U.S., the estimated number of new melanoma cases will decrease by 5–6%. The percentage of deaths expected in 2023 is 4.4%. The new estimated diagnosed cases of melanoma in the U.S. during 2023 will be 186,680. Moreover, the number of deaths in 2023 will be 7990, which have increased by 27% more than last year’s figures [6]. The typical age to be affected by this cancer is younger than 40; mainly women. It is hard to cure if it has spread to the other parts of the human body [7,8]. However, the early-stage diagnosis of melanoma can be treated quickly and has a good recovery rate [6]. Several techniques have been implemented to assist with diagnosis.

Conventional skin cancer diagnosis techniques entail a thorough process that includes a physical examination, medical history-based evaluation, dermatoscopy, imaging examination, and a pathology report. Several approaches have been introduced in the literature to differentiate malignant and benign skin lesions such as the ABCD rule [9], seven-points checklist, and three-point checklist [10]. The ABCD rule of dermatoscopy characterizes the geometrical and organizational lesion properties. The three-point and seven-point approaches identify melanoma and BCCs based on three and seven characteristics [9]. All these steps end with patient treatment [11]. In addition, the consumption of more time, costs, locations, and healthcare providers are other factors that can delay the diagnosis. Therefore, it is important to diagnose skin cancer early, which can help decrease the mortality rate and increase the survival percentage. Hence, an automated computer-aided diagnosis (CAD) system is required to accurately and efficiently diagnose skin cancer from dermatoscopy images [12]. The dermatoscope is a new non-invasive diagnosis tool for skin diseases, but it depends on the expertise of the person (doctor). Therefore, employing the concept of artificial intelligence (CAD) shows the success of addressing the above problems [13]. The AI-based CAD system can be useful at home or abroad to recognize skin cancer from the dermatoscopy images [14]. 

Early CAD systems of skin cancer were based on traditional features [15] such as texture, shape, and color; however, due to increased training images, these techniques fail to provide better results [16]. In addition, the CAD systems based on the handcrafted features faced several challenges, such as similarity in lesion shape, color, and texture, as shown in Figure 1 [17]. With the advancement of deep learning, AI-based computerized techniques show much greater success in medical imaging (detection and recognition) [18,19].

In medical imaging, the convolutional neural network (CNN) shows improved recognition performance [15]. By employing the deep backbone of CNN, the deeper layer is selected for the deep feature extraction [21]. Much research has been conducted in this domain in the last couple of years incorporating deep learning methods [22]. Despite this, many challenges still exist in this domain, including low-contrast infected lesions, variations in the shape of lesions, similarities in the colors of different skin lesion classes, imbalanced skin classes, and a few more. Based on these challenges, there is room to enhance lesion detection and multiclass classification accuracy. Hence, in this article, the following challenges are addressed: (i) imbalanced skin classes increase the probability rate of a higher number of image classes that impact the prediction performance of other classes; (ii) the low-contrast skin lesions impact the lesion localization accuracy; (iii) variations in lesion shape and texture may segment the incorrect region that later extracted the irrelevant features (incorrect region features, healthy region features, and extra features that are not required for the classification purpose). In addition, multiclass skin lesions have a high similarity in shape, color, and appearance; therefore, it is also difficult to recognize a true class correctly.

Major Contributions: Our major contributions are as follows: Proposal of a hybrid contrast enhancement technique using the fusion of top–bottom filtering and haze reduction technique.Fine-tuning of three pre-trained CNN architectures and training using transfer learning. For the training of deep learning models, a genetic algorithm is employed for the selection of hyperparameters instead of manual selection.Proposal of a serial-controlled positive correlation approach for the fusion of trained neural nets feature.Development of an improved optimization algorithm named Antlion for the feature selection.

The manuscript is organized so that Section 2 describes the related work based on skin lesion approaches. Section 3 describes proposed methodology, followed by Section 4, which elaborates on and discusses the experimental setup, results, and comparisons with existing methods. Finally, the conclusion is given in Section 5.

## 2. Related Work

It has been extensively investigated how to automatically diagnose skin cancer [23,24]. Deep learning algorithms show significant success in the area of medical imaging, especially for the identification of skin cancer [25]. The main components of traditional automated skin cancer diagnosis approaches are developing handcrafted features and using machine learning classifiers for classification [26]. A CAD system consists of a few important steps, such as preprocessing of original dermoscopy images, lesion detection using segmentation techniques, handcrafted feature extraction, feature selection, and classification using machine learning classifiers. Recently, CNNs that can learn hierarchical features have had considerable success with medical image processing, especially for skin cancer recognition [27].

Kassem et al. [28] discussed the importance of deep learning for the classification of skin cancer using deep learning techniques. They discussed extensively the importance of deep learning for better skin lesion classification, the complexity of deep learning techniques, and the most current stage of development. Hauser et al. [29] presented an explainable AI framework for skin lesion diagnosis. Zhang et al. [30] presented an attention mechanism CNN model for skin lesion recognition. Each attention block jointly used residual learning to improve representation learning. The experiments were conducted on the ISIC2017 dataset and showed improved recognition accuracy. Anand et al. [31] presented a U-NET and CNN architecture fusion for skin lesion detection and classification. They used U-NET architecture to detect lesions from the input dermoscopy images; however, CNN architecture was employed for the classification. The HAM10000 dataset was employed to validate the proposed framework and obtained accuracy above 97%. Fayadh et al. [32] introduced a wavelet transform and CNN-based architecture to diagnose skin lesions. The unwanted information was removed by employing the concept of wavelet and max pooling. Then, a residual neural network is proposed and features are extracted by employing the concept of transfer learning. The extracted features are classified using an ELM classifier and obtained improved accuracy on ISIC2017 and HAM10000 datasets.

Simon et al. [33] provided an interpretable deep-learning framework for skin lesion segmentation and classification. The main strength of this work was categorizing the tissues into 12 dermatological classes. After that, they trained a deep CNN using these characteristics for final classification. They tested the introduced framework on dermatoscopy images and compared it with clinical accuracy. During the comparison phase, the clinical method achieved an accuracy of 93.6%, whereas the computerized method attained 97.9%. This shows that the computerized methods have better performance than the clinical techniques. Javeria et al. [34] introduced an integrated model of preprocessing, segmentation, feature extraction, and deep feature fusion. Firstly, they resized the images and converted RGB into a luminance channel, then they used the Otsu algorithm and biorthogonal 2-D wavelet transform to segment the affected part of the skin. After that, pre-trained AlexNet and VGG16 were used to extract the deep features. Then, the optimal feature set was obtained using PCA for further classification. Al-Masni et al. [35] devised an integrated diagnostic paradigm encompassing skin lesions’ segmentation and classification. Inception-v3, ResNet-50, Inception-ResNet-v2, and DenseNet-201 were deployed in the DL FRCN framework using dermatoscopic images to segment regions of interest, followed by classifier over segmentation results. The proposed integrated DL model works acceptably on different types of skin lesions. The model was evaluated on a balanced, segmented, and augmented dataset, including the International Skin Imaging Collaboration (ISIC) and its variants in 2016, 2017, and 2018. Overall weighted prediction accuracy for Inception-v3, ResNet-50, Inception-ResNet-v2, and DenseNet-201 classifiers is 77.04%, 79.95%, 81.79%, and 81.27% for two ISIC2016 classes, 81.29%, 81.57%, 81.34%, and 73.44% for three ISIC2017, as well as 88.05%, 89.28%, 87.74%, and 88.70 for four ISIC2018 classes. Pacheco et al. [36] used the thirteen best deep-learning networks. Finally, they concluded that the SE Net convolutional neural network and Adam optimization were the perfect architecture among all neural networks. The proposed model obtained 91% performance on the ISIC2019 dataset. Farooq et al. [37] introduced a model to enhance the classification performance by up to 86% by incorporating Mobile Net and Inception Net. For these models, Kaggle’s updated dataset of skin cancer was utilized to check their performances. Esteva et al. [38] conducted a pioneering CNN-based research work to detect and classify skin lesion datasets. Lui et al. [39] defined a deep learning model with Dense Net and Resnet using the MFL module. The proposed work generated an effective accuracy of 87% on the ISIC2017 database for skin lesion classification. Pedro et al. [40] proposed a Feedforward Neural Network (FNN) classification model and Linear SVM on the dermo fit dataset. Their setup produced an accuracy level of 90% on the selected dataset. Milton et al. [41] depicted a comprehensive study of multiple deep-learning techniques for skin cancer. They conducted the experiments on the publicly available ISIC2018 dataset, fed to multiple neural networks, including Inception Resnet-V2, PNASNet-5, SENet-154, and Inception-V4. The PNASNet-5 model is the best performer at 76% accuracy level.

Khatib et al. [42] presented Resnet-101 architecture for the skin lesions classification. They fine-tuned the architecture by employing transfer learning (TL) to differentiate the various forms of skin lesions and achieved an accuracy level of 90% on a well-known PH2 database. Alizadeh et al. [43] deployed the Vgg19 NN model using kernel principal components analysis (KPCA) and attained 85.2% accuracy using the ISIC2016 dataset. Almaraz et al. [44] used the ABCD rule-based technique after extracting handcrafted features’ color, shapes, and texture. These features were then given to Mobile NetV2 neural network melanoma categorization. The proposed technique achieved 92.4% accuracy using the HAM10000 dataset. Reis et al. [45] employed a DL approach for skin lesion identification and segmentation. The suggested technique was investigated on three widely accessible datasets, ISIC2018, ISIC2019, and ISIC2020, where prediction accuracy was enhanced to 90.1, 90.2, and 91.3%. Khan et al. [8] presented an improved subdivision combinatorial architecture (IMFO) consisting of moth+flame and DL Classification for skin lesion classification. Furthermore, they extended the model to minimize the time taken in diagnosing skin cancer. The IMFO architecture was tested on PH2, ISBI 2016, 2017, and 2018 datasets and obtained an accuracy level of 98.70%, 95.38%, 95.79%, and 92.69%, respectively. The architecture was also tested on the dataset Ham10000, where it reflected a precision level of 90.67% which represents an improvement. Khan et al. [46] presented another intelligent system based on deep neural networks for complex skin cancer categories. The authors suggested a two-stream DNN information fusion framework for classifying multiclass skin cancer. Firstly, a contrast enhancement technique based on fusion was suggested in which magnified images were fed to the pre-formed DenseNet201 architecture. These features were modified utilizing the skewness-controlled moth + flame optimization approach. After that, stream deep features were captured and down-sampled using fine-tuned MobileNetV2 pre-trained systems and a proposed feature selection structure. The proposed technique was tested on three unbalanced datasets named as HAM10000, ISBI2018, and ISIC2019, that produced accuracy levels of 96.5%, 98%, and 89%, respectively. These discussed methods focused on detection and classification using deep learning and machine learning classifiers. They did not focus on the fusion of different source features. Also, they ignore the process of best feature selection that can help in reducing the computational time. To address these important challenges, a new AI-based fully automated framework is proposed for skin lesion classification. 

## 3. Proposed Methodology

The proposed methodology is illustrated in Figure 2. Figure 2 reflects that firstly, the dataset is preprocessed, and then the enhanced dataset is fed to fine-tune the DL model for training based on transfer learning to extract deep features. Secondly, the extracted features are passed through the feature fusion process. Finally, an updated Antlion optimization approach was employed to obtain an optimized feature vector. 

### 3.1. Datasets Description

This paper uses two variants of ISIC datasets, including 2018 and 2019, for the experimental process. 

ISIC2018: This dataset was generated in the year 2018 by ISIC. It is a collection of 10,014 training images and 55,834 testing images. The dermoscopy technology is employed for capturing images RGB images. This dataset has seven classes: Akiec, Bcc, Bkl, Df, Mel, Nv, and Vasc. Table 1 summarizes and highlights the overall class distribution within the dataset.

ISIC2019: This dataset was generated in the year 2019 by ISIC. It is a collection of 20,685 training images and 47,514 testing images. The dermatoscopy technology is employed for capturing images RGB images. This dataset has seven classes: AK, BCC, BKL, DF, MEL, NV, and VASC. Table 2 summarizes and highlights the overall class distribution within the dataset.

### 3.2. Novelty 1: Lesion Enhancement

In this work, a hybrid technique is employed for contrast enhancement. In the first step, a haze reduction technique is employed, where the input image is refined, followed by applying top–bottom filtering to improve local and global contrast [47]. The step-wise haze reduction process is given below. 

Step 1: The haze image model is given below:(1)$$I\left(x\right)=J\left(x\right)T\left(x\right)+L\left(1-T\left(x\right)\right)$$
where I, J, L, and T represent the intensity, scene radiance, atmospheric light, and map transmission, respectively. The scene radiance is recovered using the algorithm [48]; however, other factors, including J from the estimated light of the atmosphere and the map transmission, are computed as follows:(2)$$J\left(x\right)=\left(I\left(x\right)-A\right)∕\left(max\left(t\left(x\right),t0\right)\right)+A$$

Step 2: Consider λ(x,y) is an input image of dimension N×M×K where N=M=256 and  K=3. Let, ${\tilde{\lambda }}_{nz}\left(x,y\right)$ determine the haze reduction image having the same dimensions. The top hat filtering is proposed and computed using the following mathematical formulation:(3)$$\lambda_{Top}(a,b)=\left(\lambda (a,b)\circ s\right)-\lambda (a,b)$$
(4)$$\lambda_{Bot}\left(a,b\right)=\left(\lambda \left(a,b\right)\circ s\right)-\lambda \left(a,b\right)$$
(5)$$\tilde{\lambda }\left(a,b\right)=\sum_{i=1}\left(\lambda_{Top},\lambda_{Bot}\right)-\lambda_{Bot}\left(a,b\right)$$
(6)$$T=Max\left(\tilde{\lambda }\left(a,b\right)\right)$$
(7)$$F=\left\{\begin{array}{c} \tilde{\lambda }\left(a,b\right)\;\;\;\;\;\;\;\;\;for\;\;\;\tilde{\lambda }\left(a,b\right)\;\geq T\\ \lambda_{los}\left(a,b\right)\;\;\;\;\;\;\;for\;\;\tilde{\lambda }\left(a,b\right)<T \end{array}\right.$$

The visual output of this process is illustrated in Figure 3.

### 3.3. Data Augmentation

This is a process in which the data/data points are artificially increased using the existing data for better training, identification, and classification in the later stages. The advantage of data augmentation is that it improves model learning by providing a huge amount of data. Also, the cost of operations related to data collection will be reduced. The detail given in Table 2 and Table 3 shows that the total number of original images are 20,685. Before data augmentation, the contrast of the real images is improved using the proposed contrast-enhanced technique. After applying augmentation, the selected datasets were updated and are shown in Table 3 and Table 4. A few sample augmented images are illustrated in Figure 4.

### 3.4. Modified Models

In this work, different DL models were fine-tuned to obtain high-performance accuracy. These are explained in detail:

Fine-Tuned DarkNet19: The model is fine-tuned by eliminating linked, softmax, classification, and final four average-pool layers. The original model is shown in Figure 5. It is all because it is pre-trained on 1000 classes belonging to the ImageNet dataset. Hence, during the fine-tuning process, four new layers are added, including the average-pooling 2D-layer, fully connected layer, softmax layer, and classification layer. The Darknet19 model is trained through transfer learning. In the training process, several hyper-parameters are adjusted, i.e., the learning rate is 0.001, the minimum-batch size is 20, the momentum is 0.07, the optimizer is stochastic gradient descent, and the maximum epochs are 100. Finally, the trained model extracts features adopting the gap layer.

Fine-Tuned ResNet18: The ResNet18 DL model consists of 18 layers. The architecture has a fully connected combination of softmax, convolutional, pooling, and classification layers. This model uses a pooling layer named ‘pool5’ for feature extraction. From the ImageNet dataset, more than a million images will be trained on the network when you load the pre-trained version. The architecture of ResNet18 is depicted in Figure 6. The last four layers, termed the average—layer, are deleted during the fine-tuning phase, along with the fully connected, softmax, and classification layers. The previous fully connected layer was trained on an ImageNet dataset with 1000 item types.

Furthermore, four more layers are added in a fine-tuning process. These are average-pooling 2D-layer, fully connected-layer, softmax layer, and classification layer. The ResNet18 model is trained using TL. Numerous hyper-parameters are initialized and adjusted during training, such as learning rate to 0.001, mini-batch size to 20, momentum to 0.07, stochastic gradient descent optimizer, and a maximum number of epochs to 100. Finally, the trained model extracts features from the pool5 layer. 

Fine-tuned InceptionV3: The InceptionV3 DL model consists of 48 layers. The architecture contains a fully connected combination of softmax, convolutional, pooling, and classification layers. A pooling layer named the ‘avg-pool layer’ was used for feature extraction. This model was previously trained on over one million photos in the ImageNet dataset. This model is mostly used for image recognition and has a 78.1% accuracy rate. The architecture of the fine-tuned InceptionV3 is depicted in Figure 7. The last four average pool layers, along with the fully connected, softmax, and classification layers, are deleted during the fine-tuning phase. The previous fully connected-layer was trained on an ImageNet dataset with 1000 item types. Next, in a fine-tuning procedure, four new layers are added. These are the average-pooling 2D-layer, a fully connected layer, a softmax layer, and a classification layer followed by TL to train the Inceptionv3 model. Numerous hyper-parameters are initialized and adjusted during the training process, such as learning rate to 0.001, minimum batch size to 20, momentum to 0.07, optimizer of stochastic gradient descent, and maximum number of epochs to 100. Finally, the trained model is used to extract features from the avg-pool-layer.

**Transfer Learning:** In this section, TL [50] is discussed for this work. The domain, denoted by $\;F=\left\{Z,R(Z)\right\}$, is made up of two parts, i.e., a feature space Z and a marginal probability distribution  R(Z), whereas $Z=\left\{z|z_{i}\in Z,i=1,\cdots ,M\right\}\;$ and M is a dataset containing M occurrences. 

After that, the task is defined; when presented with a particular domain F, the task is represented as $T=\left\{W,f(.)\right\}\;$ including two factors such as label-space *W* and a mapping function f(.), whereby $\;W=\left\{w|w_{i}\in W,i=1,\cdots ,M\right\}$, and M is a label set for the relevant instances in F. The mapping function f(.), generally known as $\;f\left(z\right)=R\left(w|z\right)$, is a non-linear indirect function that could bridge the gap between the anticipated judgment derived from the proposed datasets and the input instance. The label spaces between these tasks also allow for the specification of different goals. Different fault classes and categories might be conceived of as distinct tasks.

Transfer learning, supplied with a source domain $F^{s}=\left\{Z^{s},R^{s}(Z^{s})\right\}$ with the source task $T^{s}=\left\{W^{s},f^{s}(.)\right\}$ and a target domain $F^{T}=\left\{Z^{T},R^{T}(Z^{T})\right\}$ with the target task $\;T^{T}=\left\{W^{T},f^{T}(.)\right\}$, is looking for a better mapping function $f^{T}(.)$ for the target task $T^{T}$ utilizing transferable knowledge from the source domain $D^{s}$ and task$\;T^{s}$. Unlike traditional ML and DL, where the domain and job of the source and target situations are identical, i.e., $F^{s}=F^{T}$ and $T^{s}=T^{T}$, TL solves challenges where the source and destination situations’ domains and/or tasks diverge, i.e., $F^{s}\neq F^{T}$ and/or $T^{s}\neq T^{T}$. 

Deep TL may be defined as follows based on the above concept: Deep TL aims to comprehend the mapping function $f^{S\to T}(.)$ Given a transfer learning challenge, leverage the sophisticated DL model that is DNN $f^{S\to T}(.)$: $Z^{T}\to W^{T}$ based on [$F^{S}$,$F^{T}$,$T^{S}$,$T^{T}$]. 

Proposed Work Process: The process of transfer learning for feature extraction of this work is depicted in Figure 8. Al three selected fine-tuned models are trained on the skin datasets using the concept of transfer learning. Deep features are extracted from the global average pooling layer of each model and obtained different dimensional feature vectors. During the training of the deep models, the hyperparameters such as learning rate, momentum, L2RegularizationFactor, and mini-batch size are selected through GA. The resultant values are given in the above section. The extracted features are further fused using a novel fusion technique (presented in the next Section 3.5). 

### 3.5. Novelty: Features Fusion and Optimization

Deep extracted features are fused using a serial correlation-based approach in this work. The main purpose of this approach is to first serially fuse all the features and then find the correlation based on the pairs. A total of four steps were performed for the fusion of this approach: 

Serially fused all vectors, as shown in Figure 3

Obtained a combined vector of dimension N×K

Find the correlation of each row feature vector and consider the most highly correlated features

Check the fitness of each row using the Fine-KNN classifier

In the end, the positively correlated and weakly correlated features are again serially fused in separate vectors. Both vectors are analyzed in terms of fitness function and the best one with better accuracy. This complete process is defined under the following Algorithm 1:
**Algorithm 1.** Input: Original feature vectorsϕ1← Fine-tune DarkNet featuresϕ2← Fine-tune Resnet18 featuresϕ3← Fine-tune InceptionV3 featuresStep 1: Fused all vectors in a serial-based fashion            ϕ4=ϕ1ϕ2ϕ3(N×k1+N×k2+N×k3)Step 2: Make sets of ϕ4 using 2×2 window size. Step 3: Find the correlation of each set using the following equation:r=n∑ϕiϕj−∑ϕi∑ϕjn∑ ϕi2−∑ϕi2n∑ ϕj2−∑ϕj2
Step 4: Consider features of positive correlation in a feature vector ϕ5k and weak correlation in ϕ6kStep 7: Fuse ϕ5k and ϕ6k separately in two new feature vectors and find the fitness of each. Step 8: Based on the fitness, consider the highest accuracy feature set for further process.Output: Positive correlation vector (higher accuracy value in this work) ←ϕ5k

The fused feature vector is further refined using a nature-inspired improved algorithm.

Antlion Optimization with Mean Deviation(ALO-MD).

Mirjalili [36] developed a novel enacted optimization approach called antlion optimization (ALO). The ALO algorithm is constructed around the inherent hunting mechanism of ant lions. 

Motivation: Antlions (doodlebugs) are classified as Myrmeleontidae and Neuroptera [51]. They often hunt as larvae, while the adult stage is used for reproduction. As they dive deep into the sand, antlion larvae move in a circular motion and spew sand from their large lips. After excavating the trap, larvae sleep under the cone’s bottom, waiting for bugs, particularly ants, to be entrapped in it. The antlion attempts to seize any prey it discovers in the snare.

On the other hand, insects try to avoid captivity and are occasionally not immediately captured. Antlions expertly pour sand towards the hole’s edge, enabling the prey to sink to the bottom. A victim trapped in the mouth is eaten underneath. Antlions fling the victim’s remains outside the hole after devouring the victim and prepare the hole for their subsequent hunt. A further interesting aspect of antlion conduct is the relationship involving trap size and two variables: hunger level and moon shape.

Antlion optimization (ALO): Mirjalili [36] developed a novel enacted optimization approach called antlion optimization (ALO). The ALO algorithm is constructed around the inherent hunting mechanism of antlions. 


**Artificial Antlion**


Using the prior depiction of antlions, Mirjalili devised the following criterion throughout optimization:Ants, as prey, wander across the search space utilizing various random walks.Antlion traps influence random walks.Antlions may dig holes in accordance to their size. The greater the fitness, the larger the hole.Antlions are more likely to capture ants if their holes are wider.An antlion with the highest fitness level in each cycle can catch any ant.The random walk’s span is adaptively reduced to simulate ants sliding toward antlions.


**Input**


A searchable area, a fitness feature, a quantity of ants, antlions, iterations, and antlions (T)


**Output**


The fitness of the elitist antlion:Make an irregular population of n ant positions and n antlion positionsDetermine the fitness of each ant and antlion.Find the elite that is the finest antlion.t = 0while(t ≤ Τ)


**for each Ant I, do**


Choose an opponent using a roulette wheel (making trap).Bring the ants nearer to the antlion; considering Equations (2) and (3).For this Ant I, build and balance a random walk; check Equations (5) and (6) for model trapping, Equation (7) for the random walk, and Equation (9) for walk normalization.end

6.Evaluate each ant’s fitness7.If an antlion grows fitter (catching prey), replace it with its equivalent ant $\left(7\right).$8.If an antlion becomes fitter than the elite, update it.9.end while


**Method 1: Antlion Optimization Algorithm (ALO)**


If an ant grows stronger than an antlion, the antlion will grab it and drag it beneath the sand.After each hunt, an antlion repositions itself near the most recently caught prey and digs a hole to maximize its chances of catching new prey.Under the conditions above, an antlion optimizer can be built in the following.**Method 1**.

**Building trap:** The hunting skill of antlions is modeled using a roulette wheel. Ants are believed to be restricted to a single antlion. The ALO algorithm must select antlions throughout optimization depending on their fitness using a roulette wheel operator. This technique increases the likelihood of stronger antlions catching ants.


**Catching prey and re-building the hole:**


In the final step of the hunt, the antlion consumes the ant. It is thought that when ants increase physical fitness in relation to their comparable antlion, they penetrate the sand and attempt to catch prey. An antlion must modify its posture to match the latest whereabouts of the chasing ant in order to maximize its potential for finding new victims. In this sense, sentence (1) is proposed.

Antliontj = Antti is better than f(antilonti), $\left(1\right)$ where t indicates the most recent revision, Antliontj represents the position of the choose j−th antlion at t−th iteration, and Antti represents the location of the I−th Ant at the t−th iteration.

Antlion optimizer, according to the algorithm, performs the following stages on each particular ant:


**Sliding ants towards Antlion:**


Sand is thrown from the hole’s center when an antlion finds an ant inside the trap. The imprisoned ant’s attempt to escape is impacted by this action. The radius of the ants’ random walk hyper-sphere is reduced adaptively to represent this behavior numerically; see Equations (8)–(10).
(8)$$a^{s}=\frac{a^{s}}{I},$$
where $a^{s}$ is the component that has the least impact on t−th iteration and i is a ratio.
(9)$$b^{s}=\frac{b^{s}}{I},$$
where $b^{s}$ is the highest value for all variables at t−th iteration and I is a ratio that is defined as:(10)$$I=10^{u}\frac{s}{S},$$
where s is the latest iteration; S the highest number of iterations; and u a constant specified by the current iteration $\left(u=2\;for\;s>0.1S,\;u=3\;otherwise\right)$. When s exceeds 0.5S, w equals 4. When s>0.75S and u=5, when s>0.9S, u=6, and when s>0.95S, u=6. Essentially, the constant u can vary the amount of exploitation precision.


**Trapping in Antlion’s holes**


The slide ant is captured by simulating the food movement towards the targeted antlion’s hole. Alternatively, the location of the selected antlion now determines how far the ant can travel. Adjusting the range of the ant’s random journey to the antlion’s location in five equations can be depicted using Equations (11) and (12):(11)asi=as+Antlionsj
(12)bsi=bs+Antlionsj
where $a^{s}$ is the least significant variable at the t−th iteration; $b^{s}$ is the vector containing all variables with the highest values at the t−th iteration; asi is the least significant factor for the i−th ant; bsj is the maximum of all variables for the i−th ant; and Antlionsj shows the location of the chosen j−th antlion at the t−th iteration.


**Random walks of ants:**


Equation $\left(13\right)$ underpins all random walks.
(13)$$y\left(s\right)=\left[0,\;cumsum\left(2p\left(s_{1}\right)-1\right);cumsum\left(2p\left(s_{2}\right)-1\right);\ldots ;cumsum\left(2p\left(s_{S}\right)-1\right)\right]$$
where the cumulative amount is calculated by cumsum; $\left(S\right)$ is the maximum number of iterations, where iteration here refers to the random walk step; and $p\left(s\right)$ is Equation (14) for a stochastic function (8).
(14)$$p\left(s\right)=\left\{\begin{array}{c} 1\;if\;rand>0.5\\ 0\;if\;rand\leq 0.5, \end{array}\right.$$
where s is the random walk step iteration in this research and rand is a random integer produced with a homogenous distribution in the range $\left[0,1\right]$ as per Equation (15):(15)Vsi=vsi−xi×(bi−asi)zsi−xi+xi
where $x_{i}$ is the random walk in the least of the i−th variable; $b_{i}$ is the random walk’s maximum value in the i−th variable; asi is the lowest of the i−th variable at the t−th iteration; and bsi is the peak of the i−th variable at the t−th iteration.


**Elitism**


The best solution(s) should be maintained throughout iterations by employing elitism. The chosen antlion and the elite antlion lead the ant’s random walk in this scenario; therefore, moving a given ant takes the form of the average of both random walks; see Equation (16).
(16)Antsi=PsA+PsF2
where PsA is the picked antlion’s random stroll about the roulette table, and PsF is the shambling around the roulette wheel of the elite antlion.

## 4. Results and Discussion

With an emphasis on the inefficiency of other classifiers, test design, data collection, recall value, quantitative data, graphical representations, and tables, this section will analyze and show the findings based on various performance indicators.

### 4.1. Experimental Setup

On the dataset, 10-fold cross-validation was used to perform the calculations. The training rate is set to 0.05, the mini-batch range is restricted to 32, and 100 iterations are required for CNN architecture learning. The best among them is validated based on performance measurements such as accuracy, time taken, sensitivity rate, precision rate, number of observations, FNR, Fowlkes–Mallows index, and F1-Score. Several classifiers are utilized to validate the suggested approach with the greatest accuracy and minimum time consumed. MATLAB 2022a was employed to execute the simulation studies on a personal desktop pc Core-i7 having a memory of 16GB as well as an 8 Gigabyte graphics card.

### 4.2. Results and Analysis

**ISIC2018 Dataset Results:** Table 5 contains the classification outcomes for the ISIC2018 dataset using the DarkNet19 deep model. The fine-tuned model was trained using the enhanced dataset, which was also used to extract features from the second-to-last feature layer. Several classifiers were used, but Quadratic SVM outperformed them with an accuracy of 86.3%, a recall rate of 87.27%, a precision rate of 87.2%, F1 score of 87.24%, and an AUC value of 0.98%. Each classifier’s computational time is also calculated, as shown in Table 5. The Fine Tree classifier’s least recorded time is 108.46 s, while the Medium Neural Network’s greatest recorded time is 2978.1 (s).

The classification outcomes of the ISIC2018 dataset for the Resnet18 deep model are shown in Table 5
**(second half)**. Numerous classifiers have been used for the classification process but Quadratic SVM performed better, achieving an accuracy of 88.3%, a recall rate of 89.39%, a precision rate of 89.13%, an F1 score of 89.26%, and an AUC value of 0.98%. Moreover, the computational time is also computed for each classifier, as shown in Table 5. Compared with experiment 1 (Table 5), it is observed that the maximum accuracy for this experiment is 88.3%, whereas for the first experiment, the maximum obtained accuracy was 86.3%. Hence, it can be summarized that the fine-tuned Resnet18 model gives better accuracy. The least noted time is 52.616 s for the Fine Tree classifier, whereas the maximum observed time is 1112.6 s for the medium neural network. 

The classification outcomes for the ISIC2018 dataset for the InceptionV3 deep model are shown in Table 5 **(third section)**. Although several classifiers were used, Quadratic SVM outperformed them all with an accuracy of 90.9%, a recall rate of 92.63%, a precision rate of 92.03%, an F1 score of 92.32%, and an AUC value of 0.99%. Each classifier’s computing time is also calculated, as shown in Table 5. It was found that the maximum accuracy for this experiment is 90.9%, compared to experiments 1 and 2. Meanwhile, for the first experiment, the maximum obtained accuracy was 86.3%, and for the second experiment, the maximum accuracy was 88.3%. Hence, it can be summarized that the fine-tuned InceptionV3 model provides better accuracy. The minimum noted time is 129.68 s for the Fine Tree classifier, whereas the maximum observed time is 4601.7 s for the medium neural network.

The classification outcomes of the proposed fusion technique on the enhanced ISIC2018 skin dataset are given in Table 6. Many classifiers were used; however, Quadratic SVM outperformed them all with an accuracy of 96.1%, a recall rate of 96.93%, a precision rate of 96.33%, an F1 score of 96.62%, and an AUC value of 0.98. Each classifier’s computational time is also calculated, as shown in Table 6. Compared with previous experiments (Table 5), it is observed that the maximum accuracy for this experiment is 96.1%, whereas for the first experiment, the maximum obtained accuracy was 86.3%, for the second experiment, the maximum accuracy was 88.3%, and for the third experiment the maximum accuracy was 90.9%. Hence, the fusion process increases accuracy more than individual deep model components. The minimum noted time is 290.756 s for the Fine Tree classifier, whereas the maximum observed time is 8693 s for the medium neural network. The confusion matrix of this experiment is also shown in Figure 9. 

Table 7 shows the proposed feature selection technique results using the enhanced ISIC2018 dataset. The Quadratic SVM classifier outperformed them all with an accuracy of 96.0%, a recall rate of 96.86%, a precision rate of 96.3%, an F1 score of 96.56%, and an AUC value of 0.99%. Each classifier’s computational time is also calculated, as shown in Table 7. Moreover, the confusion matrix is illustrated in Figure 10, which shows the correct prediction rate for each class. The highest accuracy for this experiment is 96.0%, compared with experiments 1, 2, and 3 (Table 5). Although the maximum accuracy for the first experiment was 86.3%, the maximum accuracy for the second experiment was 88.3%, the maximum accuracy for the third experiment was 90.9%, and the maximum accuracy for the fourth experiment was 96.1%.

In conclusion, it can be said that when comparing Table 5, it is shown that the optimization time increases accuracy and decreases computing time; however, it can be seen that the accuracy only changed a little, but the computational time changed significantly compared to the previous experiment. Hence, overall, the proposed framework and the optimization process show improvement. The least noted time is 130.94 s for the Fine Tree classifier, whereas the maximum observed time is 2525.7 (s) for medium KNN.

**ISIC2019 Dataset Results:** The classification outcomes of the ISIC2019 dataset using the DarkNet19 deep model are shown in Table 8. The fine-tuned model was trained using the supplemented dataset, which was also used to extract features from the second-to-last feature layer. Weighted KNN outperformed other classifiers used for classification, achieving an accuracy of 99.7%, a recall rate of 99.73%, a precision rate of 99.71%, an F1 score of 99.72%, and an AUC value of 1.00%. Each classifier’s computing time is also calculated, as shown in Table 8. The Fine Tree classifier’s minimum noted time is 245.51 s, whereas the bi-layer neural network’s highest recorded time is 2123.6 (s). 

The classification outcomes of the ISIC2019 dataset for the Resnet18 deep model are shown in Table 8 (second half). Several classifiers have been used; however, Weighted KNN outperformed them all with an accuracy of 99.5%, a recall rate of 99.53%, a precision rate of 99.59%, an F1 score of 99.56%, and an AUC value of 1.00%. Each classifier’s processing time is also calculated. This experiment obtained a maximum accuracy of 99.5% compared to the previous experiment. The classification outcomes of the ISIC2019 dataset for the InceptionV3 deep model are shown in Table 8
**(third section)**. Many classifiers have been used; however, Weighted KNN outperformed them all with an accuracy of 99.7%, a recall rate of 99.66%, a precision rate of 99.69%, an F1 score of 99.36%, and an AUC value of 1.00%. Overall, this experiment’s performance is better than previous experiments. 

The classification outcomes for the enhanced ISIC2018 skin dataset are given in Table 9. Many classifiers have been used; however, Medium KNN outperformed them all with an accuracy of 99.9%, a recall rate of 99.86%, a precision rate of 99.88%, an F1 score of 99.88%, and an AUC value of 1.00%. Each classifier’s processing time is also calculated, as shown in Table 9. Moreover, Figure 11 shows the Medium KNN’s confusion matrix to verify the correct prediction rate. Compared with the previous three experiments of the proposed fusion process, it is observed that the accuracy of this experiment is significantly improved. After the fusion process, we employed the proposed feature selection technique. 

Several classifiers have been used; however, Weighted KNN outperformed them all with an accuracy of 99.9%, a recall rate of 99.89%, a precision rate of 99.89%, an F1 score of 99.88%, and an AUC value of 1.00%. Each classifier’s computing time is also calculated, as shown in Table 10. Moreover, Figure 12 also shows the Weighted KNN confusion matrix. By employing Figure 12, we can verify the correct prediction rate of each cancer class. In contrast to Experiment 1, Experiment 2, Experiment 3, and Experiment 4 (Table 8 and Table 9), it is noted that the maximum accuracy for this experiment is 99.9%. In contrast, the maximum accuracy for the first experiment was 99.7%, the maximum accuracy for the second experiment was 99.5%, the maximum accuracy for the third experiment was 99.7%, and the maximum accuracy for the fourth experiment was 99.9%. Overall, it is noted that the accuracy of the fusion process is improved, but computational time is significantly reduced for the feature selection technique. 

In the end, the comparison is conducted regarding time for the middle steps on selected datasets. Table 11 presents the computational time-based comparison of the ISIC2018 dataset. This table shows that the time noted by the Resnet18 model is less than the Darknet19 and InceptionV3, except for the Bagged Tree classifier. However, after the fusion process, it jumped and almost doubled this time, which is a drawback of this framework. This drawback was resolved through a proposed optimization approach that maintains accuracy and reduces the computational time significantly compared to the fusion process. For Darknet19, the minimum time is 108.46 s for the Fine Tree classifier, and the maximum time is 2978.7 s for the Medium Neural Network. For Resnet18, the minimum time is 52.616 s for the Fine Tree classifier, and the maximum time is 1112.6 s for the Medium Neural Network. For InceptionV3, the minimum time is 129.68 s for the Fine Tree classifier, and the maximum time is 4601.7 s for the Medium Neural Network. For fusion, the minimum time is 290.756 s for the Fine Tree classifier, and the maximum is 8693 s for the Medium Neural Network. For optimization, the minimum time is 130.94 s for the Fine Tree classifier, and the maximum time is 2525.7 s for the Medium KNN.

Table 12 presents the computational time-based comparison of the ISIC2019 dataset. This table shows that the time noted by the Resnet18 model is less than the Darknet19 and InceptionV3 except for Medium KNN, Weighted KNN, Medium Neural Network, and Bi-Layered Neural Network classifier. However, after the fusion process, it jumped and almost doubled this time, which is a drawback of this framework. This drawback was resolved through a proposed optimization approach that maintains accuracy and reduces the computational time significantly compared to the fusion process. For Darknet19, the minimum time is 245.51 s for the Fine Tree classifier, and the maximum time is 8373 s for Bagged Tree. For Resnet18, the minimum time is 102.59 s for the Fine Tree classifier, and the maximum time is 7982.8 s for Bagged Tree. For InceptionV3, the minimum time is 460.25 s for the Fine Tree classifier, and the maximum time is 5605.7 s for the Medium Neural Network. For fusion, the minimum time is 460.25 s for the Fine Tree classifier, and the maximum is 13082.3 s for the Bi-Layered Neural Network. For optimization, the minimum time is 33.252 s for the Fine Tree classifier, and the maximum is 1018.1 s for Bagged Tree. Finally, the proposed framework’s accuracy is compared with several recent studies, as presented in Table 13. Based on this table, it is observed that the proposed framework accuracy is significantly improved. In addition, a few AI-based dermatoscopy techniques (publicly available) are compared with the proposed method. In [52], they obtained an AUC value of 0.970 on ISIC2019 and 0.932 on ISIC2018 dataset using the ADAE technique. However, our method obtained 0.99. In [53], they obtained an accuracy of 96.10%, whereas the proposed method obtained 99.8%.

Table 14 presents the summary of all best results based on the additional performance measures such as Fowlkes–Mallows index, MCC, and Kappa. Overall, the proposed method shows the improved accuracy. 

## 5. Conclusions

Today, serious issues include the deaths of patients due to the late or incorrect diagnosis of cancer cases. Early diagnosis of cancer cases using a CAD system can help in the reduction in the death rate. When an appropriate CAD system is employed, this can complement the work of dermatologists in classifying skin lesions (benign or melanoma). This work proposes a deep learning- and optimization-based end-to-end framework for multiclass skin lesion classification. Initially, a contrast enhancement technique was proposed based on the dark channel haze and top–bottom filtering that improved image quality and the strength of deep features. Hyperparameters of the fine-tuned model were initialized using a genetic algorithm instead of manual initialization. After that, deep features were extracted and fused with the information using a serial correlation approach. The fusion process improved the accuracy, but computational time increased. A selection technique called improved antlion optimization was developed to make the framework more efficient in terms of time. The best features are selected using this approach and classified using machine learning classifiers. The experimental process was conducted on two publicly available datasets, ISIC2018 and ISIC2019, and obtained improved accuracy of 96.1% and 99.9%, respectively.

### 5.1. Limitations

-A detailed analysis is required for the max pooling operation of sizes 2 × 2, 3 × 3, and 4 × 4 of the weights preprocessing process.-The augmentation process improved the accuracy, but on the other hand, it significantly increased the redundant features.-KNN classifiers drop the classification accuracy that needs the proper analysis.-The fusion process improved the accuracy, but computational time also increased due to the enlarged number of predictors.

### 5.2. Future Directions

A residual block-based attention network will be designed in the future, and more layers will be added based on the GradCAM approach. This will allow max-pooling layer weights to be analyzed to help improve the proposed model. In addition, the experimental process will be conducted on the ISIC2020 dataset. 

## Figures and Tables

**Figure 1 diagnostics-13-02869-f001:**
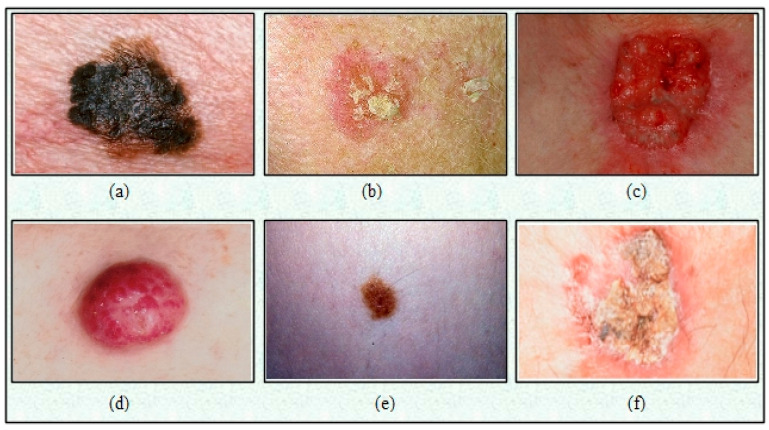
Categories of skin lesions: (**a**) melanoma; (**b**) actinic keratosis; (**c**) basal cell carcinoma; (**d**) Merkel cell carcinoma; (**e**) melanocytic nevus/mole; (**f**) squamous cell carcinoma [20].

**Figure 2 diagnostics-13-02869-f002:**
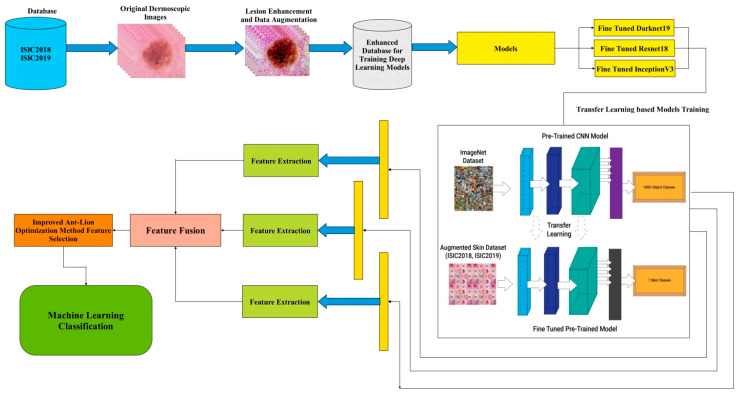
Illustration of the proposed methodology for skin lesion classification.

**Figure 3 diagnostics-13-02869-f003:**
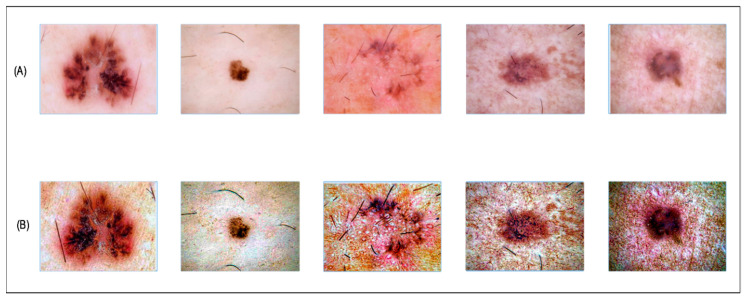
Visual results of hybrid contrast enhancement. (**A**) Original images; (**B**) enhanced images after applying the proposed approach.

**Figure 4 diagnostics-13-02869-f004:**
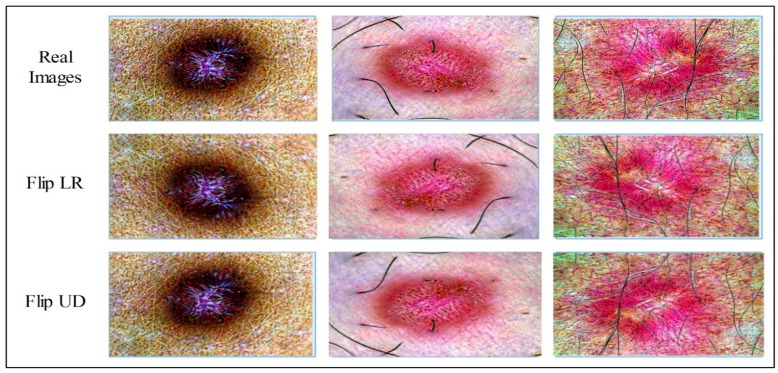
Augmentation process such as Flip LR and Flip UD.

**Figure 5 diagnostics-13-02869-f005:**
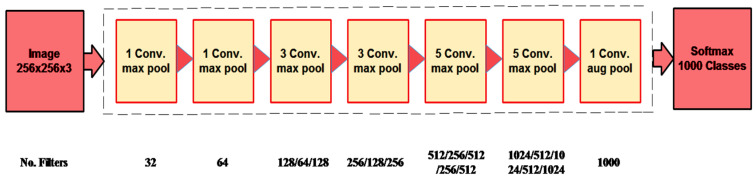
Architecture of DarkNet19 model.

**Figure 6 diagnostics-13-02869-f006:**
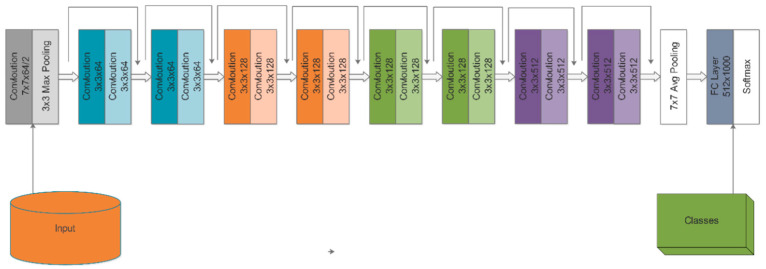
Architecture of ResNet18 model [49].

**Figure 7 diagnostics-13-02869-f007:**
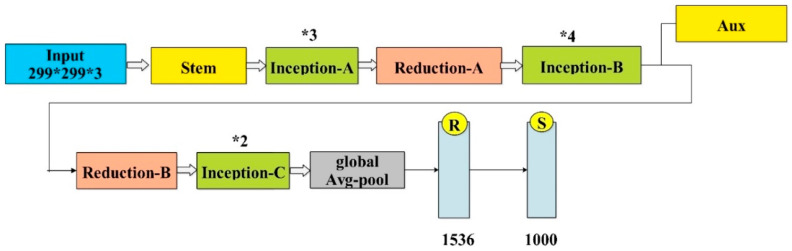
Architecture of InceptionV3 model [49]. * sign in employed for such kind of figures.

**Figure 8 diagnostics-13-02869-f008:**
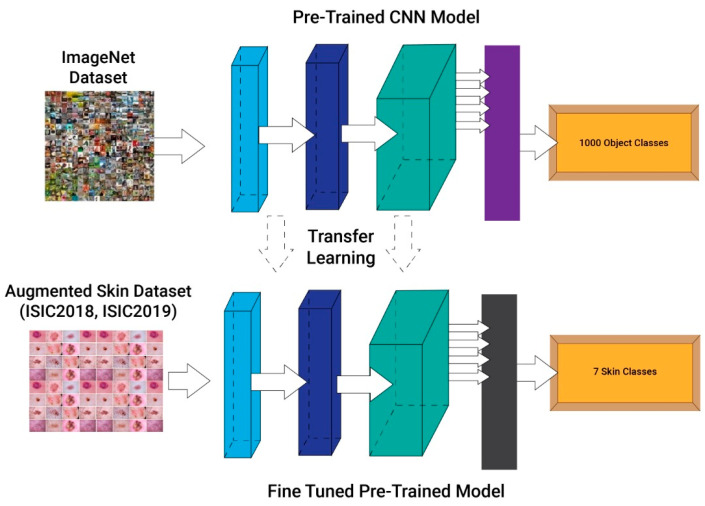
Process of transfer learning for feature extraction of skin cancer.

**Figure 9 diagnostics-13-02869-f009:**
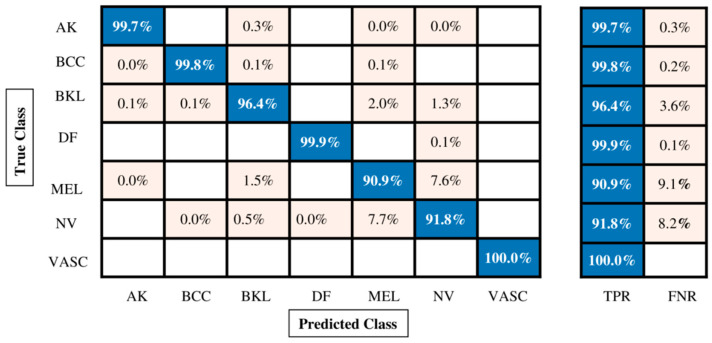
Confusion matrix of Quadratic SVM for augmented ISIC2018 dataset.

**Figure 10 diagnostics-13-02869-f010:**
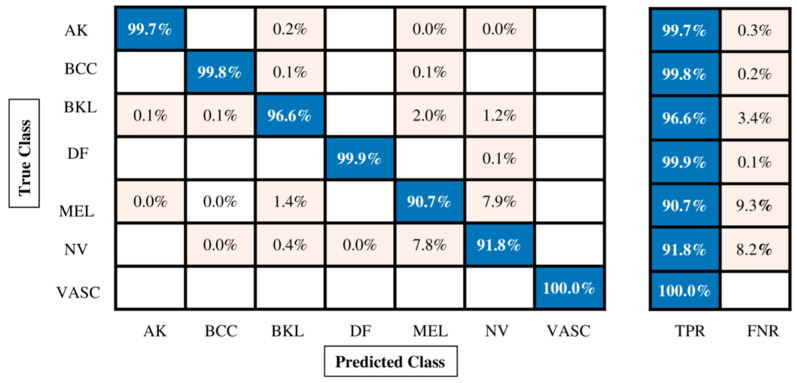
Confusion matrix of Quadratic SVM for augmented ISIC2018 dataset.

**Figure 11 diagnostics-13-02869-f011:**
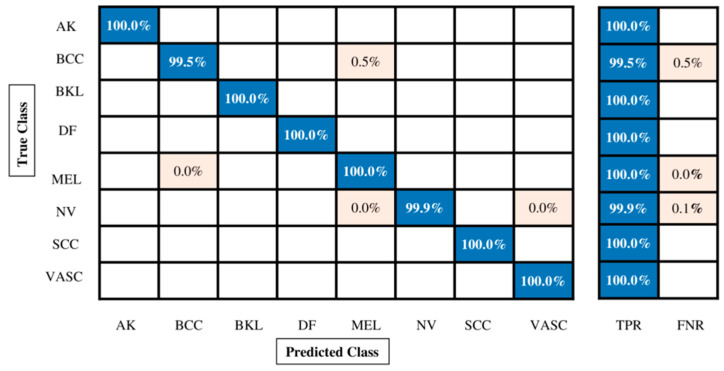
Confusion matrix of Quadratic SVM for augmented ISIC2019 dataset.

**Figure 12 diagnostics-13-02869-f012:**
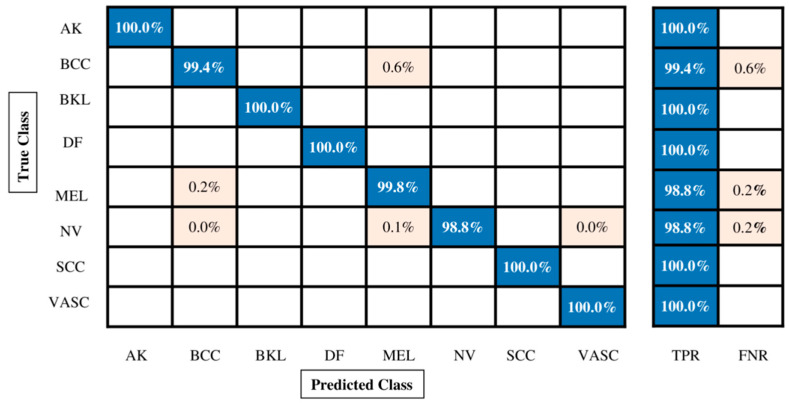
Confusion matrix of Quadratic SVM for augmented ISIC2019 dataset using proposed feature selection technique.

**Table 1 diagnostics-13-02869-t001:** ISIC2018 Skin dataset description.

Class	No. of Images
Akiec (Actinic keratosis)	326
Bcc (Basal Cell Carcinoma)	514
Bkl (Benign keratosis)	1099
Df (Dermatofibroma)	115
Mel (Melanoma)	1113
Nv (Nevus)	6705
Vasc (Vascular)	142

**Table 2 diagnostics-13-02869-t002:** ISIC2019 Skin dataset description.

Class	No. of Images
AK (Actinic keratosis)	3469
BCC (Basal Cell Carcinoma)	3232
BKL (Benign keratosis)	3200
DF (Dermatofibroma)	3232
MEL (Melanoma)	3072
NV (Nevus)	2112
SCC (Squamous cell carcinoma)	3200
VASC (Vascular)	2240

**Table 3 diagnostics-13-02869-t003:** Updated ISIC2018 Skin dataset images after data augmentation.

Class	Before Augmentation	After Augmentation
Akiec (Actinic keratosis)	326	7821
Bcc (Basal Cell Carcinoma)	514	7201
Bkl (Benign keratosis)	1099	6593
Df (Dermatofibroma)	115	7360
Mel (Melanoma)	1113	6678
Nv (Nevus)	6705	15,637
Vasc (Vascular)	142	4544

**Table 4 diagnostics-13-02869-t004:** Updated ISIC2019 Skin dataset after data augmentation.

Class	Before Augmentation	After Augmentation
AK (Actinic keratosis)	3469	6938
BCC (Basal Cell Carcinoma)	3232	6464
BKL (Benign keratosis)	3200	6400
DF (Dermatofibroma)	3232	6464
MEL (Melanoma)	3072	6144
NV (Nevus)	2112	4224
SCC ()	3200	6400
VASC (Vascular)	2240	4480

**Table 5 diagnostics-13-02869-t005:** Classification results of fine-tuned darknet19 and ResNet18 models on augmented ISIC2018 skin dataset.

Classifier	Classification Results of Fine-Tuned Darknet19 Model on Augmented ISIC2018 Skin Dataset						
	Sensitivity (%)	Precision Rate (%)	F1 Score (%)	Area Under Curve	Fowlkes–Mallows Index	Accuracy (%)	Time (s)
Fine Tree	56.53	58.8	57.64	0.84	57.65	59.1	108.46
Quadratic SVM	87.27	87.2	87.24	0.98	87.23	**86.3**	2740.8
Medium KNN	83.84	81.79	82.81	0.98	82.81	82.5	1923.7
Weighted KNN	86.91	85.41	86.154	0.96	86.16	85	2657.1
Bagged Tree	80.84	82.2	81.52	0.96	81.52	81.0	604.7
Narrow Neural Network	81.21	80.24	79.74	0.94	80.72	79.5	2701.3
Medium Neural Network	84.3	83.89	84.1	0.95	84.09	82.8	2978.7
Bi-Layered Neural Network	81.66	80.71	81.18	0.94	81.18	79.9	2809.8
**Classifier**	**Classification Results of Fine-Tuned Resnet18 Model on Augmented ISIC2018 Skin Dataset**						
	**Sensitivity** **(%)**	**Precision Rate (%)**	**F1 Score (%)**	**Area Under Curve**	**Fowlkes–Mallows Index**	**Accuracy** **(%)**	**Time** **(s)**
Fine Tree	61.91	62.69	62.3	0.87	62.30	62.9	52.616
Quadratic SVM	89.39	89.13	89.26	0.98	89.26	**88.3**	868.55
Medium KNN	87.17	86.01	86.58	0.98	86.59	85.4	429.95
Weighted KNN	89.3	88.39	88.84	0.97	88.84	87.4	428.64
Bagged Tree	83.76	84.37	84.06	0.97	84.06	83.5	145.55
Narrow Neural Network	85.46	84.56	85	0.96	85.01	83.8	985.98
Medium Neural Network	85.84	85.63	85.74	0.98	85.73	84.5	1112.6
Bi-Layered Neural Network	85.3	84.51	84.9	0.96	84.90	83.6	1065
**Classifier**	**Classification Results of Fine-Tuned inceptionV3 Model on Augmented ISIC2018 Skin Dataset**						
	**Sensitivity** **(%)**	**Precision Rate (%)**	**F1 Score (%)**	**Area Under Curve**	**Fowlkes–Mallows Index**	**Accuracy** **(%)**	**Time** **(s)**
Fine Tree	89.614	89.04	89.32	0.98	89.33	87.9	129.68
Quadratic SVM	92.63	92.03	92.32	0.99	92.33	**90.9**	2425.5
Medium KNN	92.14	90.99	91.56	0.99	91.56	90.0	1780.6
Weighted KNN	91.21	90.54	90.88	0.97	90.87	89.2	1872.2
Bagged Tree	90.53	90.29	90.4	0.98	90.41	89.1	314.74
Narrow Neural Network	90.89	90.57	90.72	0.99	90.73	89.3	3566.7
Medium Neural Network	89.99	89.99	89.98	0.99	89.99	88.6	4601.7
Bi-Layered Neural Network	91.34	90.74	91.04	0.99	91.04	89.5	3565.7

**Table 6 diagnostics-13-02869-t006:** Classification results of fusion on augmented ISIC2018 skin dataset.

Classifier	Sensitivity (%)	Precision Rate (%)	F1 Score (%)	Area Under Curve	Fowlkes–Mallows Index	Accuracy (%)	Time (s)
Fine Tree	91.03	90.41	90.72	0.98	90.72	89.8	290.756
Quadratic SVM	96.93	96.33	96.62	0.98	96.63	**96.1**	6034.85
Medium KNN	95.69	94.44	95.06	0.99	95.06	93.8	4134.25
Weighted KNN	96.24	94.94	95.58	0.99	95.59	94.3	4957.94
Bagged Tree	94.24	93.93	94.08	0.99	94.08	93.5	1064.99
Narrow Neural Network	95.2	95.07	95.14	0,98	95.13	94.6	7253.98
Medium Neural Network	95.5	95.31	95.4	0.99	95.40	94.8	8693
Bi-Layered Neural Network	95.04	94.9	94.96	0.99	94.97	94.4	7440.5

**Table 7 diagnostics-13-02869-t007:** Classification results of proposed optimization algorithm on augmented ISIC2018 skin dataset.

Classifier	Sensitivity Rate (%)	Precision Rate (%)	F1-Score (%)	Area Under Curve	Fowlkes–Mallows Index	Accuracy (%)	Time (s)
Fine Tree	90.67	90.21	90.44	0.98	90.44	89.7	130.94
Quadratic SVM	96.92	96.35	96.64	0.99	96.63	**96.1**	1464.4
Medium KNN	95.79	94.47	95.12	0.99	95.13	93.9	2525.7
Weighted KNN	96.33	94.99	95.66	0.99	95.66	94.4	2120
Bagged Tree	93.91	93.66	93.78	0.99	93.78	93.2	268.03
Narrow Neural Network	94.57	94.6	94.58	0.98	94.58	94.0	963.51
Medium Neural Network	95.16	94.97	95.06	0.99	95.06	94.5	430.84
Bi-Layered Neural Network	94.41	94.33	94.36	0.98	94.37	93.8	1562.8

**Table 8 diagnostics-13-02869-t008:** Classification results of the ISIC2019 skin dataset.

Classifier	Sensitivity (%)	Precision Rate (%)	F1 Score (%)	Area Under Curve	Accuracy (%)	Fowlkes–Mallows Index	Time (s)
Fine Tree	80.95	81.06	81.0	0.96	80.5	81.00	245.51
Quadratic SVM	99.34	99.36	99.34	1.00	99.3	99.35	547.26
Medium KNN	99.19	99.24	99.22	1.00	99.2	99.21	1951.8
Weighted KNN	99.73	99.71	99.72	1.00	**99.7**	99.72	1916.6
Bagged Tree	99.11	99.2	99.16	1.00	99.2	99.15	8373
Narrow Neural Network	99.59	99.6	99.58	1.00	99.6	99.59	1793.8
Medium Neural Network	99.61	99.61	99.6	1.00	99.6	99.61	3073
Bi-Layered Neural Network	99.5	99.54	99.52	1.00	99.5	99.52	2123.6
**Classifier**	**Classification Results of Fine-Tuned Resnet18 Model on Augmented ISIC2019 Skin Dataset**						
	**Sensitivity** **(%)**	**Precision Rate (%)**	**F1 Score (%)**	**Area Under Curve**	**Accuracy** **(%)**	**Fowlkes–Mallows Index**	**Time** **(s)**
Fine Tree	63.78	66.54	65.14	0.88	63.9	65.15	102.59
Quadratic SVM	98.53	98.6	98.56	1.00	98.5	98.56	466.99
Medium KNN	98.51	98.81	98.66	1.00	98.7	98.66	1675.7
Weighted KNN	99.53	99.59	99.56	1.00	**99.5**	99.56	1612.6
Bagged Tree	96.89	96.89	96.88	1.00	97.2	96.89	7982.8
Narrow Neural Network	98.26	98.35	98.3	0.99	98.3	98.30	2934.5
Medium Neural Network	98.34	98.45	98.4	0.99	98.4	98.39	4256.6
Bi-Layered Neural Network	98.29	98.4	98.34	0.99	98.4	98.34	6263.8
**Classifier**	**Classification Results of Fine-Tuned inceptionV3 model on augmented ISIC2019 Skin Dataset**						
	**Sensitivity** **(%)**	**Precision Rate (%)**	**F1 Score (%)**	**Area Under Curve**	**Accuracy** **(%)**	**Fowlkes–Mallows Index**	**Time** **(s)**
Fine Tree	96.2	96.26	96.22	0.99	96.3	96.23	112.15
Quadratic SVM	99.19	99.24	99.22	1.00	99.2	99.21	595.26
Medium KNN	99.05	88.61	93.54	1.00	99.0	93.68	1348.2
Weighted KNN	99.66	99.69	99.68	1.00	**99.7**	99.67	1361.8
Bagged Tree	99.35	99.39	99.36	1.00	99.4	99.37	4002.9
Narrow Neural Network	98.49	99.51	98.98	0.98	99.5	99.00	4175.2
Medium Neural Network	99.49	99.5	99.5	0.98	99.5	99.49	5605.7
Bi-Layered Neural Network	99.33	99.34	99.36	1.00	99.3	99.33	4694.9

**Table 9 diagnostics-13-02869-t009:** Classification results of the proposed fusion on augmented ISIC2019 skin dataset.

Classifier	Sensitivity (%)	Precision Rate (%)	F1 Score (%)	Area Under Curve	Accuracy (%)	Fowlkes–Mallows Index	Time (s)
Fine Tree	95.99	96.05	96.02	0.99	96.1	96.02	460.25
Quadratic SVM	99.54	99.54	99.56	1.00	99.6	99.54	1609.51
Medium KNN	99.86	99.88	99.88	1.00	99.9	99.87	4875.7
Weighted KNN	99.93	99.94	99.94	1.00	99.9	**99.93**	4891
Bagged Tree	99.38	99.44	99.42	1.00	99.4	99.41	8903.5
Narrow Neural Network	99.76	99.78	99.76	1.00	99.8	99.77	8903.5
Medium Neural Network	99.83	99.84	99.84	1.00	99.8	99.83	12,935.3
Bi-Layered Neural Network	99.79	99.79	99.78	1.00	99.8	99.79	13,082.3

**Table 10 diagnostics-13-02869-t010:** Classification results of proposed optimization algorithm on augmented ISIC2019 skin dataset.

Classifiers	Sensitivity (%)	Precision Rate (%)	F1 Score (%)	Area Under Curve	Accuracy (%)	Fowlkes–Mallows Index	Time (s)
Fine Tree	98.29	94.8	96.52	0.99	94.6	96.53	33.252
Quadratic SVM	99.6	99.6	99.6	1.00	99.6	99.60	142.73
Medium KNN	99.8	99.84	99.82	1.00	99.8	99.82	279.97
Weighted KNN	99.89	99.89	99.88	1.00	99.9	**99.89**	283.46
Bagged Tree	98.78	98.85	98.82	1.00	98.8	98.81	1018.1
Narrow Neural Network	99.61	99.65	99.64	1.00	99.6	99.63	61.565
Medium Neural Network	99.73	99.71	99.72	1.00	99.7	99.72	62.441
Bi-Layered Neural Network	99.59	99.59	99.58	1.00	99.6	99.59	83.467

**Table 11 diagnostics-13-02869-t011:** Computational time-based comparison for ISIC2018 skin dataset.

Classifier	Darknet19	Resnet18	InceptionV3	Fusion	Optimization
Fine Tree	108.46	52.616	129.68	290.756	130.94
Quadratic SVM	2740.8	868.55	2425.5	6034.85	1464.4
Medium KNN	1923.7	429.95	1780.6	4134.25	2525.7
Weighted KNN	2657.1	428.64	1872.2	4957.94	2120
Bagged Tree	604.7	145.55	314.74	1064.99	268.03
Narrow Neural Network	2701.3	985.98	3566.7	7253.98	963.51
Medium Neural Network	2978.7	1112.6	4601.7	8693	430.84
Bi-Layered Neural Network	2809.8	1065	3565.7	7440.5	1562.8

**Table 12 diagnostics-13-02869-t012:** Computational time-based comparison for the ISIC2019 skin dataset.

Classifier	Darknet19	Resnet18	InceptionV3	Fusion	Optimization
Fine Tree	245.51	102.59	112.15	460.25	**33.252**
Quadratic SVM	547.26	466.99	595.26	1609.51	**142.73**
Medium KNN	1951.8	1675.7	1348.2	4875.7	**279.97**
Weighted KNN	1916.6	1612.6	1361.8	4891	**283.46**
Bagged Tree	8373	7982.8	4002.9	8903.5	**1018.1**
Narrow Neural Network	1793.8	2934.5	4175.2	8903.5	**61.565**
Medium Neural Network	3073	4256.6	5605.7	12,935.3	**62.441**
Bi-Layered Neural Network	2123.6	6263.8	4694.9	13,082.3	**83.467**

**Table 13 diagnostics-13-02869-t013:** Comparison of the proposed framework with recent computerized AI techniques.

Authors/Reference	Method	Dataset		Accuracy (%)	Time (s)
		ISIC18	ISIC19		
Nawaz, Marriam [54]	A deep learning CornerNet and Fuzzy-Means Clustering Algorithm	**✓**		99.63%	
Alsaade [55]	Deep Learning and Traditional Machine learning based AI system	**✓**		98.35%	
Babu [56]	Support vector machine and HOG features-based AI system	**✓**		76%	
Alizadeh [57]	Combining CNN and Traditional Features of AI System		**✓**	97.5%	
Ichim [58]	Multiple Connected Neural Network Architecture		**✓**	97.5%	
El-Khatib [42]	Simple Deep Learning Method		**✓**	93%	
Monika [59]	Machine learning-based system		**✓**	96.25%	
**Our Proposed System**		**✓**		**96.1%**	1464.4
**Our Proposed System**			**✓**	**99.9**	283.46

**Table 14 diagnostics-13-02869-t014:** Proposed classification results for ISIC2018 and ISIC2019 datasets based on all performance measures including MCC, Kappa, and Fowlkes–Mallows index.

Classification results of fine-tuned darknet19 model on augmented ISIC2018 skin dataset								
Quadratic SVM	Sensitivity (%)	Precision Rate (%)	F1 Score (%)	Area Under Curve	Fowlkes–Mallows index	Accuracy (%)	MCC	Kappa
	87.27	87.2	87.24	0.98	87.23	**86.3**	84.93	44.33
Classification results of fine-tuned resnet18 model on augmented ISIC2018 skin dataset								
Quadratic SVM	89.39	89.13	89.26	0.98	89.26	**88.3**	87.16	51.92
Classification results of fine-tuned inceptionV3 model on augmented ISIC2018 skin dataset								
Quadratic SVM	92.63	92.03	92.32	0.99	92.33	**90.9**	90.44	61.92
Classification results of fusion on augmented ISIC2018 skin dataset.								
Quadratic SVM	96.93	96.33	96.62	0.98	96.63	**96.1**	95.93	84.09
Classification results of proposed optimization algorithm on augmented ISIC2018 skin dataset.								
Quadratic SVM	96.92	96.35	96.64	0.99	96.63	**96.1**	95.94	84.10
**Proposed classification results for ISIC2019 dataset**								
Classification results of fine-tuned darknet19 model on augmented ISIC2019 skin dataset								
Weighted KNN	Recall (%)	Precision Rate (%)	F1 Score (%)	Area Under Curve	Accuracy (%)	Fowlkes–Mallows index	MCC	Kappa
	99.73	99.71	99.72	1.00	99.7	99.73	99.68	98.63
Classification results of fine-tuned Resnet18 model on augmented ISIC2019 skin dataset								
Weighted KNN	99.53	99.59	99.56	1.00	99.5	99.53	99.45	97.78
Classification results of fine-tuned InceptionV3 model on augmented ISIC2019 skin dataset								
Weighted KNN	99.66	99.69	99.68	1.00	99.7	99.66	99.62	98.37
Classification results of fusion on augmented ISIC2019 skin dataset.								
Weighted KNN	99.93	99.94	99.94	1.00	99.9	99.93	99.91	99.63
Classification results of proposed optimization algorithm on augmented ISIC2019 skin dataset.								
Weighted KNN	99.89	99.89	99.88	1.00	99.9	99.89	99.90	99.60

## Data Availability

The datasets used in this work are publicly available.

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
