# Peer review of "SkinNet-INIO: Multiclass Skin Lesion Localization and Classification Using Fusion-Assisted Deep Neural Networks and Improved Nature-Inspired Optimization Algorithm"

_diagnostics, 2023, doi:10.3390/diagnostics13182869_

Round 1

Reviewer 1 Report

I may only comment on the dermato-oncologic aspects of the manuscript.

1) Throughout the whole introduction, the description of melanoma is insufficient and sometimes wrong and the used references are not relevant. The clinical aspects should be completely rewritten by a dermato-oncologist using appropriate references.

2) The authors should use common metrics such as sensitivity, specifity, positive and negative predictive value.

3) The authors should compare the performance of their algorithm with open source AI algorithms for melanoma such as ADAE on the two original data sets and then focus on the cases which are not correctly classified and try to extract out of this comparison a statement which is of interest to clinicians.

Author Response

Dear reviewer thank you very much for given us an opportunity to improve this version. We tried our best to improve this version. thanks

Reviewer 2 Report

The abstract needs quantification. Need for this research has to be specified. Section 1 and 2 had 15 references each why? section 2 needs modification and improvement. Why Bio inspired algorithms are introduced suddenly in the paper. MCC, Kappa and Error rate are to be included. Table 7,8, and 9 why fine tree exhibits poor performance? how do you tackled the data imbalance problem. why accuracy is almost 99%  for all the classifiers. More information is required for transfer learning of the models. The conclusion requires modifications.

NIL

Author Response

Dear Reviewer, thank you for your valuable comments. Response sheet is attached. Thanks

Round 2

Reviewer 1 Report

1) The authors have not involved a dermatologist for their statements on malignant melanoma.

Just a few erroneous statements:

Malignant Melanoma is a rare cancer type, and the survival rate of Melanoma is less than 5%.

Melanosomes are the precursors to skin cancer.

However, these methods are time-consuming, require much attention, availability of a dermatologist, and are costly.

Traditional clinical approaches for melanoma diagnosis are unsuccessful nowadays.

Doctors' manual detection of cancer cases (naked eye) is time-consuming, costly, and lacks expertise.

2) The obtained results have not been put into perspective with studies using AI for dermatoscopy performed by dermatologist. The manuscript only compares results to publications of other bioinformatic scientists.

Author Response

Dear honorable reviewer, we done the essential changes as per your recommendation. For each comment, a response is provided. thanks

C1: The authors have not involved a dermatologist for their statements on malignant melanoma.

Response: Dear honourable reviewer, thank you for your valuable comments. We are computer scientists and working on the publically available skin cancer datasets such as ISIC2018, ISIC2019, and HAM10000. In our group and region, there are not any dermatologists; that is a reason, we did not added dermatologists.  

C2: Just a few erroneous statements:

Malignant Melanoma is a rare cancer type, and the survival rate of Melanoma is less than 5%.

Melanosomes are the precursors to skin cancer.

However, these methods are time-consuming, require much attention, availability of a dermatologist, and are costly.

Traditional clinical approaches for melanoma diagnosis are unsuccessful nowadays.

Doctors' manual detection of cancer cases (naked eye) is time-consuming, costly, and lacks expertise.

Response: As per recommendation and keen observations by honourable reviewer, we updated our introduction section with more important information and corrected the above statements. Thank you

C3: The obtained results have not been put into perspective with studies using AI for dermatoscopy performed by dermatologist. The manuscript only compares results to publications of other bioinformatic scientists.

Response:  Dear reviewer, thank you for your valuable comments.

Please accept our apology in this comments, as we are expert in the computerized techniques which are based on the deep learning (like comparison table). We don’t know how to compare the AI for dermoscopy techniques. However, I see a few like [1] and [2], they used few dermoscopy techniques.

In [1], they obtained the AUC value of 0.970 on ISIC2019 and 0.932 on ISIC2018 dataset using ADAE technique. However, our method obtained 0.99.

In [2], they obtained accuracy of 96.10%, whereas the proposed method obtained 99.8%.

[1]: Prospective validation of dermoscopy-based open-source artificial intelligence for melanoma diagnosis (PROVE-AI study)

[2]:  AI Techniques of Dermoscopy Image Analysis for the Early Detection of Skin Lesions Based on Combined CNN Features

Reviewer 2 Report

All the corrections are included in the paper. Hence, there is no need for further corrections.

NIL

Author Response

Dear reviewer, thank you very much for your valuable recommendation. thank you